# Selected Organizational and Managerial Aspects of Health and Nutrition Education of Various Types of Consumers of Spa Treatment Services in Poland

**DOI:** 10.3390/nu14112337

**Published:** 2022-06-02

**Authors:** Joanna Woźniak-Holecka, Tomasz Holecki, Kajetan Suchecki, Sylwia Jaruga-Sękowska

**Affiliations:** 1Department of Health Promotion, Faculty of Health Sciences in Bytom, Medical University of Silesia in Katowice, 41902 Bytom, Poland; jwozniak@sum.edu.pl (J.W.-H.); sjaruga@sum.edu.pl (S.J.-S.); 2Department of Health Economics and Health Management, Faculty of Health Sciences in Bytom, Medical University of Silesia in Katowice, 41902 Bytom, Poland; tholecki@sum.edu.pl; 3Department of Market and Consumption, Faculty of Economics, University of Economics in Katowice, 40287 Katowice, Poland

**Keywords:** consumer education, nutritional education, spa resort, health promotion, health education

## Abstract

Consumer education, including nutrition education, understood as a process based on scientific principles, is becoming a very effective element in influencing the health of the population in the modern world. This work is based on direct research carried out in 2016–2018 in the form of a questionnaire interview among patients—consumers of educational services in spa treatment facilities. The research sample was *N* = 1000 (600 people were tested with the use of PAPI (Paper and Pencil Interview), and 400 people with the use of CAWI (Computer-Assisted Web Interview)) and concerned a representative group of spa patients in Poland. In addition, as a supplement, a direct survey was conducted in the form of a Computer-Assisted Telephone Interview with managers of spa treatment facilities (*N* = 50). Consumers of spa treatment services differ from each other, and types can be distinguished based on their professional activity status and the type of entity that pays for their stay in the spa, and, using cluster analysis, the status of professional activity in relation to their education level. The nutritional education system is based on highly specialized medical personnel. At the same time, it does not use the available assessment tools based on proven monitoring and evaluation indicators. Health education, which also includes nutritional education, apart from disease prevention, is the basic tool for health promotion in spa treatment conditions, enabling the introduction of a permanent change in the patients’ lifestyles, the additional advantages of which are low costs and relatively high efficiency. In the course of the study, several useful patient profiles were also distinguished, thanks to which it is possible to select dedicated educational methods for selected groups of recipients.

## 1. Introduction

In the modern world, consumer nutrition habits that can be positively shaped are becoming one of the important elements of influencing the health of the population or selected subpopulations. The most important tool in the catalog of real impact is health education, including nutritional education, understood as a process based on scientific principles, creating an opportunity for planned learning and aimed at enabling individuals to make informed decisions about health and act in accordance with them [1].

Eating habits and nutritional education are often indicated by researchers as one of the most important non-medical elements of health prevention in society [2] from infancy [3], and among children and adolescents [4,5,6], yet such studies are relatively less frequently conducted among the elderly (ex. Jeruszka-Bielak et al. [7]), which may result from the very process of education, which, formally, usually ends at the stage of young adulthood. The role of proper nutrition among representatives of various professional groups is also discussed [8].

Important elements in shaping eating behavior are geophysical, socio-economic, and cultural factors. The availability of particular food products may depend on the local topography, climate, and the development of trade and transport in a given area. Furthermore, the choice of food is also based on socio-economic factors, which are influenced, among others, by the level of education, family status, income, and knowledge about food and its impact on the proper functioning of the body [9]. Improper eating behavior and lack of physical activity can lead to the development of diet-related diseases such as obesity, type 2 diabetes, and cardiovascular disease. In order to prevent negative health effects resulting from improper diet, it is recommended to raise the level of society’s knowledge of food and its impact on health through health education, including nutritional education [10]. Passing on and acquiring skills necessary to shape and improve eating behaviors and dealing with previously unknown situations are inherent elements in nutritional education. It allows people to make changes to their current lifestyles [11], and also promotes health behaviors such as physical activity and balanced nutrition. Proper motivation influences the pursuit of behavioral changes that are unfavorable to health. In order to understand recipients and effectively help them change inappropriate eating behaviors, it is important to know the aspects that influence people’s choices and behavior [12].

One of the areas where it is possible to shape consumer behavior is spa treatment, which is an element of the Polish health care system [13,14]. In the past, the tasks undertaken by spa treatment facilities concerned only a narrow area of rehabilitation and treatment of chronic diseases. Currently, spa therapy also includes health promotion, disease prevention, health education, and nutritional education, understood as the conscious creation of learning opportunities, aimed at changing behavior in the area of health and nutrition [15].

Healthcare units providing health services in the field of spa treatment or spa rehabilitation under the concluded contracts are obliged to provide food adequate for the health condition and needs of patients [16]. Meals provided to patients as part of the provision of health services should not only ensure the 24 h energy and nutritional needs, but also fulfill the function supporting the treatment process, which is important for recovery [17]. Therefore, it is necessary to develop eating habits related to the adoption of the consumption model as part of a healthy diet [18,19], which is most often also associated with conscious and sustainable consumption on the food market [20].

A properly balanced diet can shorten the time of treatment and convalescence, and thus reduce the cost of total treatment by up to 30–50% [21]. In order to give the appropriate importance to the nutritional problems of patients in Poland, systemic solutions are necessary. The development of standards and requirements for nutrition in spa treatment facilities will facilitate avoiding the generally prevailing freedom in decisions when it comes to energy and nutritional value.

## 2. Materials and Methods

The methodology of the study included a comparative-descriptive analysis that allowed the pre-existing knowledge to be compared with new facts and dependencies, as well as a quantitative statistical analysis. It was based on a questionnaire survey aimed at patients who were beneficiaries of spa services, and carried out by a substantively prepared interviewer using the following methods:Paper and Pencil Interview (PAPI)—on a research sample of 600 respondents;Computer-Assisted Web Interview (CAWI)—on a research sample of 400 respondents.

The questionnaire for patients consisted of 13 questions, including 7 metric questions about demographic characteristics, professional status, and the entity referring to spa treatment and the place of treatment, as well as 6 substantive questions regarding various aspects of health education, including nutritional education, carried out during a medical stay in a spa.

The second part of data collection was based on a questionnaire survey addressed to the management of health resorts (unit managers) with the use of:Computer-Assisted Telephone Interview (CATI)—on a research sample of 50 respondents.

The questionnaire consisted of 13 questions concerning basic information about the treatment unit and the characteristics of health education for patients carried out in the unit.

The study concerned a group of spas from different areas of Poland, selected in accordance with the principles of quota selection and carried out in 2016–2018.

The majority of the surveyed population was based on elderly people who are natural beneficiaries of spa services. The dominant age group was patients aged 60–69 (37.8%), and the second-largest group were people aged 50–59 (31.1%). The older ones constituted 14.3% of the respondents, and the youngest (under 50) only 3.4% of the respondents. Less than half of the respondents had secondary education (42.3%), the next largest group were people with higher education (35%), and patients with lower education levels in total constituted 22.7%. The vast majority of respondents (64.4%) were married, 13.5% of patients were single, and 18.2% were widowers and widows.

In developing the typology of consumers [22] on the market of spa treatment services, based on the criteria of the status in the labor market and the type of entity paying for the stay in, cross tables were used, generated on the basis of the variables adopted in the IBM SPSS Statistics 25 program. The statistical significance level of α = 0.05 was adopted in the conducted analysis. Calculations were made using significance tests, determining the *p*-value test probability for individual variables. Quantitative variables are presented as percentage shares of their respective categories.

The second typology, in which the adopted criteria were level of education and labor market status, was prepared on the basis of a cluster analysis using the PS Imago Pro 7.0 software (IBM SPSS Statistics 27).

In the research questionnaire that was used in direct research, consumers were asked to provide their level of education (primary, vocational, secondary, and higher) and their status in the labor market (working, retired, disability pensioner, or unemployed).

The first element of the analyses was the decision to use one of the hierarchical agglomeration methods—the Ward method with the square of the Euclidean distance. The application of this method was carried out in several stages. The diagnostic variables (level of education and status in the labor market) were selected, and the distance matrix was calculated using a computer program. On this basis, a dendrogram was created. The obtained binary tree shows successive combinations of clusters from 1000, i.e., in a situation when each observation is in a separate cluster, up to 1, when all observations are in one cluster. The hierarchy obtained in this way allows the mutual position of clusters to be determined. The adopted cut-off axis (at level 11) allowed for the assumption of the existence of three clusters, which enabled further analyses to be carried out.

After determining the probable number of clusters, k-means cluster analysis was used. Here, too, previously identified variables (level of education and status in the labor market) were used, and an ordinal number was used to describe the observations, i.e., exactly the same data as in the case of Ward’s hierarchical method. Initial cluster centers and an ANOVA table were calculated (Table 1), and as a last resort, three clusters were obtained, the size of which ranged from 13.4% of observations to 69.1% of observations. The description of clusters distinguished in this way was preceded by an analysis of the structure of each of these clusters.

## 3. Results

The research conducted with patients in the form of an interview revealed that, regardless of the type of facility and institution that patients are referred to for spa treatment, every form of therapy includes services in the field of health education and nutritional education, which also results from the law [23]. However, this does not change the fact that there are noticeable differences in the interest in education and satisfaction with its implementation among various types of consumers of spa services, distinguished on the basis of two criteria adopted in the analysis, i.e., the criterion of professional activity (being one of the socio-economic factors), within which the professionally active people (working and unemployed) and professionally inactive (retirees and pensioners) were specified, and the criterion related to the participation of other entities in the process of purchasing spa services—consumers were distinguished as: those whose services were paid by the payer of the health care system (Polish National Health Fund—NFZ), paid by the payer of the pension and disability prevention system (Social Insurance Institution—ZUS, Agricultural Social Insurance Institution—KRUS, and State Fund for the Rehabilitation of the Disabled—PFRON (formally, it is not part of the pension and disability prevention system; however, due to the nature of activities for people with disabilities and the simplification of the typology, it was decided that it classified)), and those who paid for their stay themselves (commercial stays). Based on the criteria adopted in this way, five types of consumers of spa services have been distinguished:Economically active users of services within the healthcare system (21.7%);Economically active users of services under the disability prevention system (23.8%);Economically inactive users of services within the healthcare system (41.8%);Economically inactive users of services under the disability prevention system (6.4%);Users of services on a commercial basis (6.3%).

In the last type, economically active people constitute 2.1% of the total number of respondents, and economically inactive people constitute 4.2%. Due to the small size of these groups and a similar profile of behavior, it was decided to consider them as one type of consumer within the assumed criteria.

Professionally active people using services under the health care system relatively rarely declared that they had been offered health education services (41%). Almost 37% said that there were no such services, and one-fifth (22%) was unable to answer this question. At the same time, 17% of this type of respondent said that the lack of participation was due to a lack of interest. This was the type of consumer who was least satisfied with education during a spa stay (73%).

Professionally active users of services under the disability prevention system much more often indicated that health education services were present during their spa stay (66%). Only 21% said they were absent, and 13% were unable to answer. At the same time, they expressed less interest in such activities. This was stated by a quarter of respondents of this type. Nevertheless, as many as 88% of those participating in it were satisfied with their education.

The economically inactive using services under the health care system slightly less frequently indicated that they were offered health education services (38% compared to 39%). A quarter (23%) did not know whether such an offer existed. About 15% of people were not interested in this offer, but 83% of people of this type reported their satisfaction with the implementation.

The economically inactive using services under the disability prevention system as often (66%) as the professionally active using the services under the disability prevention system indicated that they were allowed to participate in educational classes. A quarter of participants (27%) were not interested in them, but 81% of those who benefited expressed their satisfaction.

Those using the services on a commercial basis were the least aware of whether such services were available during their stay (59%). Only 22% of the people of this type indicated that they could use them, and 19% said that there was no such possibility. Interestingly, in this type, only 8% of people said that they were not interested in such activities at all. There were also no negative assessments regarding satisfaction with participation in this type of activity, but in this case, 78% of respondents from this type did not provide an answer to this question.

The second of the proposed typologies was based on the criteria of level of education and status in the labor market. Based on the cluster analysis carried out, information was obtained about which diagnostic variables predominantly determined the creation of individual clusters, and consequently, allowed to name the types of consumers using spa treatment services obtained in this way.

In the first, the smallest, group, representing 13.4% of the respondents, there were people with different levels of education (mostly secondary and vocational) and at the same time retired or on a disability pension. This type was defined as “people with average education, not working due to their age or health condition”.

It is also the group that most often (56%) indicated that health education was provided during their stay at the health resort. Less than 14% indicated that they were not interested in this type of activities, and at the same time, every fourth person who participated in such activities indicated that they were not satisfied with them, which was the highest percentage among the surveyed clusters.

The second group included 17.5% of the respondents. It is made up of people with primary and vocational education, who had the status of an employed person or were already retired. This type was defined as “poorly educated people working now or in the past”.

This is the type of consumers who most often indicated that they did not know whether health education was provided during their stay in a spa facility (28.6%). This result is not surprising, taking into account the fact that a quarter of respondents (25%) were not interested in such activity at all. On the other hand, among those who participated in it, as many as 90% expressed their satisfaction with these activities, which was the highest result among the types discussed.

The third cluster is the most numerous cluster. Observations belonging to it constitute as much as 69.1% of the total. These are people with secondary and higher education, who at the time of the study had the status of a working person or were already retired. This type was defined as “the most educated people working now or in the past”.

These people were the least likely to confirm that no health education was conducted during their stay in the health resort (43.8%). Among the people who participated in such classes, the vast majority (83.8%) were satisfied with them.

In general, among all respondents (*N* = 1000), less than half of the respondents (45.8%) declared participation in health education classes during their stay at the sanatorium. Patients who declared that they did not participate in health education classes were asked about the reasons for their absence. Almost 2/3 of patients indicated the fact that during their stay such classes were not held at all, and only 17% of people were not interested in them. This result indicates the need to improve the rank and organization of training in the field of health education for spa patients, and it may be presumed that with an appropriate promotion of the courses they will show their willingness to participate in such activities.

The topics discussed during the classes mainly included a healthy lifestyle (76.2%), rational nutrition (71.6%), and physical activity (65.1%). Much less often the focus was on the issues of disease risk factors (43.2%), the characteristics of a given disease entity (48.7%), and pharmacotherapy (22.3%). Therefore, it should be concluded that in this respect the needs of the consumers had not been fully met, and they relate directly to the goals of spa therapy, especially pronounced in older age groups with multiple diseases. When broken down into individual payers, it is clear that in the case of the payer of the health care system, the focus was mainly on healthy lifestyle and nutritional education, whereas in the case of the payer of the pension and disability prevention system and commercial patients, the focus was primarily on a healthy lifestyle and physical activity. Details are presented in Table 2.

As the patients declared, the person who had conducted educational training most often was a medical doctor (35.2%) or a physiotherapist (20.7%). In the field of nutrition, education is conducted by a dietitian employed in the spa treatment entity (10%), which is appropriate. A public health specialist (9.8%) is only in the fourth position among the mentioned specialists while conducting health education is one of the basic tasks assigned to this profession [24,25].

Oral lectures were the preferred form of training (74.2%), which belong to the verbal methods, currently considered low-effectiveness forms of training. The training, which in principle combines theoretical knowledge with methods based on practical skills, was attended by only 10.7% of patients [26].

Only 15% of the surveyed patients declare that they completed the evaluation questionnaire after completing the health education classes, which is clear negligence regarding the professionalization of the training. The evaluation process is aimed at checking whether the education was effective, i.e., whether the set goals were achieved [27], so it is a very important element of the implementation of any type of training, enabling the assessment of existing assumptions and drawing conclusions for the future, which may contribute to the improvement of the effectiveness of learning process.

The research in the form of a telephone interview carried out among the personnel managing health resorts showed that 92% of the surveyed establishments conduct health education in their area; therefore, in subsequent questions regarding the detailed aspects of health education, only the units that declared that they performed these sentences were taken into account (*N* = 46).

All surveyed entities declared that classes in education are of a group nature, in addition, every third entity also conducts individual classes. From the point of view of the effectiveness of educational programs, it is important that classes in this field should be cyclical. Most of the surveyed units declare that there were at least three meetings with patients (63%).

An effective health education program should meet three basic conditions for implementation, which include framework, spirality, and continuity. The first relates to the educator’s freedom to select content depending on the health needs of a given community and the existing conditions and possibilities. Spirality requires the sender of educational messages to systematically return to basic topics at subsequent levels of education, along with supplementing them with further information necessary in the course of education. The latter condition refers to the necessity of cyclical implementation of classes, which, to a greater extent than one-time lectures, guarantees the formation or change of health-related attitudes and behaviors [28]. The optimal length of a single meeting is about 60 min. Most of the training sessions in health resorts lasted only 30 min (41%), which from the point of view of the effectiveness of educational activities seems to be too short a time of impact. Health education programs should be created in relation to specific groups of recipients in order to obtain high effectiveness of the activities carried out. The main issues undertaken during educational activities in the surveyed entities concerned nutritional education (78%), a healthy lifestyle (70%), physical activity (66%), and specific disease entities (64%). These results are similar to those obtained in the survey conducted among consumers of spa services. Similar to the first part of the results, the low percentage of institutions dealing with pharmacotherapy (16%) draws attention here. Health education classes in spa treatment entities are conducted mainly by the staff who have the closest contact with the patient, i.e., a medical doctor (70%) and a nurse (66%). There is a risk that the quintessentially medical personnel, due to the competencies assigned to the profession, will prefer health education based on a disease-oriented model or a risk factor-oriented model, ignoring the health-oriented model, which from the point of view of health promotion is a recommended approach that may be by far the greatest contributor to a lasting change in the behavior of consumers of health services. Only one of the surveyed institutions employs public health specialists, who often have specialist education in the indicated scope, with a diploma of “health promotion” or “health education” specialization. What is particularly worrying, however, is the tendency to outsource these tasks to external companies (24%), most often without appropriate substantive supervision. In such situations, concerns about the quality of the classes are justified.

Attention should be paid to the clear weakness of the education system in the context of broadly understood medical staff, especially the lack of professional educators. The role of the health educator is complex. Transferring only theoretical knowledge about health or disease, without taking into account a wide range of various methods and techniques aimed at improving health status, will not be effective. Practical methods, based on the active participation of patients, are considered to be the most effective methods of transferring information in the field of health education. In the surveyed institutions, the dominant form of transferring knowledge was oral lectures (80%), many entities also use materials and leaflets (24%), which is not an effective form of education that could contribute to a real change of behavior and consolidation of positive health patterns. Most of the surveyed units (84%) do not carry out the evaluation process of the educational activities carried out, which would be an opportunity to check the level of knowledge acquired by the participants of the classes, and also to verify the quality and gradually improve the procedures.

## 4. Discussion

The growing health awareness of society means that health and spa tourism is an increasingly dynamically developing branch of tourism. Although quite often these two terms are used interchangeably and their terminological scope does not fully coincide, it should be noted that health tourism is an increasingly popular form of spending time [29], and even a form of investment from the point of view of the concept of human capital. This reinforces the concept that good health is an important condition for the development of human and social capital.

The specificity of sanatorium therapy is a factor contributing to effective education processes. The patient is isolated from his or her environment, separated from everyday duties and habits, including those related to a faulty lifestyle and eating habits. They come for a stationary stay of at least three or, under certain conditions, four weeks, which guarantees a sufficiently long time of exerting a positive social impact. This also applies to the possibility of modifying the patient’s diet by developing healthy habits that can be preserved after returning from the sanatorium at the patient’s home. For the patients, it may be the only contact with a nutrition professional in their lives and a chance to learn the rules of balancing meals, healthy eating choices, other food processing techniques, or even new products in the kitchen that he has not used so far.

Nutritional well-being is a fundamental aspect of health and quality of life, especially in the elderly. It is estimated that at least half of the elderly require nutritional intervention to improve their health, and 85% have one or more chronic diseases that could be improved with proper nutrition [30].

As medical procedures are usually planned in the morning hours, the afternoon hours can be used to educate patients. The research shows that patients often complain about a lack of activities later in the day. These circumstances can be considered favorable and possible to use by organizing an attractive cycle of educational activities for patients, responding to their health needs.

Currently, the Polish healthcare system is dominated by a practice of one-way flow of information, based on the biomedical paradigm [31]. It mainly refers to the theory of the functioning of the body and the treatment of diseases, and not to the understanding of the mechanisms shaping the actions and attitudes of individuals. It is also necessary to have access to dietary help for people who need to change their eating habits due to their lifestyle or a history of illness. Improper nutrition in non-sanatorium conditions may have a negative impact on health or lead to the aggravation of the disease, and as a consequence, the effects obtained during spa treatment are weakened [32].

Therefore, an inherent element of spa treatment is initiating, supporting, and monitoring activities in the field of nutritional education, aimed at raising awareness in this area, and, if necessary, attempting to change the current eating habits. Ideas such as the reference to the values and ways of thinking of the addressees, or the non-authoritarian style of communication, currently underestimated in health and nutrition education, may determine its effectiveness, especially at the individual level [33]. Currently, multilevel approaches are being promoted in order to increase nutritional security, especially among the elderly, poor or less educated groups. A tiered approach means applying a multi-directional model of action to improve the nutritional status of the population, and includes: nutrition education materials, providing participants with healthy recipes, workshops, tasting events with healthy products, improving participants’ culinary skills in preparing low-cost meals, and changing the food policy in the area., e.g., by reducing the prices of such products or by increasing access to healthy food [34]. Even short-term nutritional education has been shown to be effective, making people more aware of the sodium content of processed foods as well as nutrient and caloric foods, the importance of exercise, and the interpretation of nutritional levels [35].

One of the methods of implementing health education in sanatoriums, indicated by health care experts, was the idea of developing training cycles compatible with the individual treatment directions of health resorts. Conducting health education and health promotion directed in accordance with the therapeutic profile is also included in the list of guaranteed services in the field of spa treatment [36].

The dominant forms of classes in spa facilities are delivery methods based on verbal communication: lecture, reading, not always enriched with visual methods, e.g., multimedia presentations, which seems to be a standard today. Meanwhile, as shown by the memory pyramid (known as Dale’s experience cone), traditional teaching methods are not sufficient to achieve satisfactory effects of changing the behavior of health service consumers. It is necessary to extend them by the activating methods, as they are commonly called, based on experience, practice, and active participation of the educated [37].

The method of disseminating health education should be based on the profession of an educator-specialist in health education or an educator-dietitian, which would allow for an increase in both the standard and scope of activities in the area of health promotion and prevention and prevention of diet-related diseases, and also improve the quality of services. Strengthening health promotion and prophylaxis may contribute to slowing down the avalanche increase in costs in this sector. The current directions of activities have focused on the implementation of these activities by medical professionals for whom conducting health education is only one of numerous, but the primary, professional tasks. Medical personnel will always put disease treatment in the first place among their duties, which is obvious, but at the same time, it is associated with relegating health education to the background.

Health educators could, on the other hand, develop and use the knowledge and skills specific to their profession, as well as significantly support treatment processes, providing, in cooperation with medical doctors and other medical professionals, extensive information support, primarily regarding further treatment and rehabilitation, but also including information of an organizational nature. When asked about this issue, experts often proposed making the educator a kind of a patient’s caregiver or a therapy coordinator who would accompany the patient during the entire stay at the spa facility, taking care of, apart from the health education, the planning of treatments and contact with other specialists employed in the facilities. The presented solutions are in line with the global tendency to delegate some tasks that do not require highly specialized competencies to other specialists. Examples include the prescription of certain medications by nurses or the initial review of some research by medical technicians. This enables a more effective allocation of human resources and, at the same time, brings considerable savings [38].

When assessing the effectiveness of classes, the periodicity of activities should also be taken into account. In order to change the behavior and health habits of patients, it is necessary to plan classes in the form of a cycle. Meanwhile, in the surveyed spa facilities, as many as a quarter of meetings were of a one-off nature, and a little over 60% of meetings were held 3 or more times. The preliminary pilot results of the empirical analysis published by E. Tomiak [39] concerning the evaluation of 2-year prophylactic studies of cardiovascular diseases in the practice of a family doctor indicate that one-time educational counseling as part of the prevention of cardiovascular diseases did not bring the expected results in the studied patients, expressed in the improvement of the values of the studied parameters with re-examination. This observation proves that health education should be a comprehensive, repetitive intervention.

In some of the surveyed spa treatment facilities, the method of providing ready-made materials was the dominant educational technique. It should be expected that this form of training will not bring any tangible results and should be abandoned. As evidenced by the analysis of training effectiveness, with a traditional allocation of resources, only 15% of participants permanently change their behavior, and 70% of people usually make unsuccessful attempts. If, on the other hand, the allocation of resources changes and more attention is devoted to preparation before the training and implementation activities after it, the effectiveness of activities may increase even up to 85% [40].

A training project that assumes a change in consumer eating habits should be assessed primarily in terms of its effectiveness. Without such an assessment, it is not possible to state whether the adopted goals have been achieved, and then whether the participants have developed their competencies to a satisfactory degree. Measurement of the effectiveness of the training project also provides information on whether the transfer of new knowledge has been made and whether the expected changes have occurred as a result of it, having a positive impact on the overall effectiveness. The evaluation of a training project is a summing-up stage that gives guidance in the human resource development planning process and is a very important, but still underrated, element of the training project. Since each activity involves a certain financial outlay, measuring the effectiveness of training projects is also a natural need to verify whether the investment in human resources was profitable [41].

It is also worth noting that educating patients in the field of a healthy lifestyle, eating habits, and pro-health activities is also associated with the institution that pays for the consumer’s stay. The organization of health education is clearly different in relation to patients for whom the governing body is the payer of the pension and disability prevention system in relation to patients directed by the payer of the health care system, which is related to the use of various measures of effectiveness, quality and evaluation of the assessed process. Health and nutrition education classes are much more likely to take place during stays financed by the institutions of the pension and disability prevention system than the health care system, and it is the health care system payer who should be particularly interested in better health education of the society, which could result in better health education in the future. the effectiveness of preventive actions. As a consequence, this would reduce public health costs, or at least use them more efficiently.

Summarizing the considerations on the theoretical and methodological foundations of health education, which should result in a change in the eating habits of consumers, it should be stated with absolute certainty that the effectiveness and efficiency of taking it lies both on the part of the sender and the recipient of the messages. In order to achieve success, the educator should take into account the personality of the recipient, the level of his knowledge and skills, education, professional and civil status of patients, and the activity itself should be carried out in an atmosphere of mutual trust and cooperation, with a focus on problem solving and a permanent change of behavior towards pro-health. However, it should not be forgotten that, according to the basic principle of health promotion, which states that every person must take responsibility for the chosen way of life and take into account the consequences of this choice for their health, the end result of the actions taken will always depend on the educated person.

The results of various empirical studies indicate the existence of a relationship between well-being, measured, e.g., by the size of GDP, and health outcomes, e.g., life expectancy [42]. As investments in health favor an increase in productivity and professional activity, they can be considered one of the main resources of the national economy [43].

## 5. Conclusions

On the basis of the conducted research, it was established that nutritional education, apart from disease prevention, is the basic tool for health promotion in spa treatment conditions, enabling the introduction of a permanent change in the way of nutrition of medical service consumers, the additional advantages of which are low costs with relatively high efficiency.

In Poland, the education system of sanatorium patients is based on medical personnel: medical doctors, nurses, and physiotherapists. The applied solution is a classic example of using highly specialized personnel for tasks that can be performed by people with lower competency potential, e.g., nutritionists or health educators.

Patients’ interest and motivation to follow the recommendations of health and nutritional education should be increased, as this will bring tangible benefits for the long-term effects of the therapy. It is necessary to take care of the form and methodology of education carried out in health resorts so that it is attractive to recipients. It is important that the education provided is multi-level and, in addition to theoretical elements, also includes workshop and practical elements related to the current market offer and taking into account the different economic and professional status and level of education of the people being educated.

It is necessary to construct the educational offer in terms of selected groups of patients because spa treatment facilities are a place for gathering patients who generate the highest health needs: the elderly and the disabled, but also people who are still professionally active and present in the labor market, which should be taken into account when planning, developing, and diversifying change management programs in terms of existing habits.

## Figures and Tables

**Table 1 nutrients-14-02337-t001:** ANOVA table for clusters made by education level and labor market status criterions.

Questions	Cluster		Error			
	Average Square	df	Average Square	df	F	Significance
Education level	203.849	2	0.277	997	736.160	<0.001
Labor market status	176.631	2	0.244	997	723.342	<0.001

Source: own study based on the obtained results.

**Table 2 nutrients-14-02337-t002:** Health education topics depending on the type of payer in the opinion of patients.

Topics of Classes	Type of Payer		
	Payer of the Health Care System	Payer of the Pension and Disability Prevention System	Commercial Stays
Physical activity	59%	63%	71%
Nutrition education	67%	45%	57%
Pharmacotherapy	18%	37%	29%
Healthy lifestyle	73%	66%	72%
Disease risk factors	38%	55%	43%
Discussion of disease entities of spa patients	41%	58%	42%

Source: own study based on the obtained results.

## Data Availability

Not applicable.

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
