# Peer review of "Selected Organizational and Managerial Aspects of Health and Nutrition Education of Various Types of Consumers of Spa Treatment Services in Poland"

_nutrients, 2022, doi:10.3390/nu14112337_

Round 1

Reviewer 1 Report

Dear authors,

I found this manuscript well-written, proper English, interesting topic, strong statistics, good results, good discussions and recent references cited, and practical conclusions. I like you emphasized the importance of Dietitians in health care teams management.

Author Response

There is no responses of reviewer.

Reviewer 2 Report

Interesting topic with valid data collection, however, the design and the interpretation have to be substantially improved.

The literature review part is very narrow and lacks a deeper analysis of international results. In the international literature, there is an abundant list of research focusing on consumers' attitudes towards nutritional values, these should be covered by the manuscript in more detail. (e.g., https://www.ncbi.nlm.nih.gov/pubmed/32882950).

The data collection, per sé, is very promising, however, the use and the interpretation of the results is very poor. Basically, the Results part in the present form is just a simple statistical description. The rich dataset would allow having a deeper analysis (at least a simple factor/cluster analysis).

The Discussion part should have more comparisons between the previously identified literature's results.

The section Conclusions should be strengthened. 

Author Response

Point 1:

The literature review part is very narrow and lacks a deeper analysis of international results. In the international literature, there is an abundant list of research focusing on consumers' attitudes towards nutritional values, these should be covered by the manuscript in more detail. (e.g., https://www.ncbi.nlm.nih.gov/pubmed/32882950).

Response 1: we added some literature review, especially during the discussion. Ex the points of literature: 28, 29, 33, 34, 41, 42

Point 2: 
The data collection, per sé, is very promising, however, the use and the interpretation of the results is very poor. Basically, the Results part in the present form is just a simple statistical description. The rich dataset would allow having a deeper analysis (at least a simple factor / cluster analysis).

Response 2:

Added a cluster analysis and new typology of consumers. Plus ANOVA table.

Point 3: The Discussion part should have more comparisons between the previously identified literature's results.

Response 3:

we expanded the section Discussion and added some points of discussion with literature no: 28, 29, 33, 34, 41, 42

Point 4: The section Conclusions should be strengthened.

Response 4: We extended the section Conclusions with points about multilevel education, importance of workshops, currently market offer education into account of differences of types of consumers.

Reviewer 3 Report

This is a survey study including 1000 participants. Various topics of classes were recorded and compared among type of payer relationship. They found that it is necessary to construct the educational offer in terms of selected groups of patients because spa treatment facilities are a place for gathering patients who generate the highest health needs.

This is an interesting study with some new findings in this area of research. The sample size of subjects is enough for analysis. However, I nevertheless have the following comments that required to be addressed.

1.      The statistical methods used and described should be made specific to the research question. Professional editing for improvement of “Statistical analysis” is suggested.

2.      For tables, I suggested to add a table about results of survey-weighted linear regression.

3.      Please add a flow chart to increase readability.

4.      Please brief the conclusions to add readability.

5.      Any study involved health belief theory for this issue?

Author Response

Point 1: The statistical methods used and described should be made specific to the research question. Professional editing for improvement of "Statistical analysis" is suggested.

Response 1: added professional describing of statistical analysis - ex. cluster analysis. It is specific for the research question of types and their differences

Point 2: For tables, I suggested to add a table about results of survey-weighted linear regression.

Response 2:Added the ANOVA table.

Point 3: Please add a flow chart to increase readability.

Response 3: According to the authors, the collected data does not allow for the presentation of a valuable flow chart.

Point 4: Please brief the conclusions to add readability.

Response 4: Added: "

It is important that the education provided is multi-level and, in addition to theoretical elements, also includes workshop and practical elements related to the current market offer and taking into account the different economic and professional status and level of education of the people being educated." - there are brief conclusions about differencies of consumer types and the consequences of these differences for their nutritional education

Point 5: Any study involved health belief theory for this issue?

Response 5: The authors did not find this type of study

Round 2

Reviewer 2 Report

The manuscript was substantially improved, however, it might still not reach the level of Nutrients, due to the shortcomings indicated earlier (poor interpretation of the results, in particular).

Reviewer 3 Report

No further comments. Thanks for your efforts on revision.

This manuscript is a resubmission of an earlier submission. The following is a list of the peer review reports and author responses from that submission.